# Barriers to and impacts of hepatitis C treatment among people who inject drugs in Kenya: A qualitative study

**Hannah N. Manley**[1], **Lindsey R. Riback**[1], **Mercy Nyakowa**[2], **Matthew J. Akiyama**[1], **Peter Cherutich**[2], **John Lizcano**[3], **Ann Kurth**[3], **Abbe Muller**[3]\*

**1** Department of Medicine, Albert Einstein College of Medicine, Montefiore Medical Center, Bronx, New York, United States of America, **2** Kenya Ministry of Health, National AIDS & STI Control Programme (NASCOP), Nairobi, Kenya, **3** Yale School of Nursing, Yale University, New Haven, Connecticut, United States of America

\* abbe.muller@gmail.com

**Data Availability Statement:** The data that support the findings of this study are available on request. The data are not publicly available due to the fact

## Abstract

Hepatitis C (HCV) disproportionately affects people who inject drugs (PWID). Despite availability of safe and effective treatment, HCV treatment access and uptake among PWID in low- and middle-income countries (LMICs) has been limited. Understanding the lived experiences of PWID in these settings who have undergone treatment provides the opportunity to gain insight into how to implement treatment programs that meet the needs of this population. Using Rhodes' Risk Environment Framework to guide our work, we conducted semi-structured interviews with 35 PWID who received HCV treatment in methadone clinics and drop-in-centers (DICs) in Nairobi and coastal Kenya supported by peer case managers from August to September 2019. Translated and transcribed interviews were analyzed thematically. Three overarching themes emerged in our thematic analysis: 1) Financial constraints as a barrier to HCV treatment, 2) HCV-related stigma, and 3) HCV treatment impacts on health and risk behaviors. These data signal unique challenges faced by PWID seeking HCV treatment in this LMIC setting and highlight the importance of interventions to reduce barriers to treatment. In order for positive treatment outcomes to be sustained, HCV treatment programs must address the barriers patients face at multiple levels and implement system-level changes.

## Introduction

Hepatitis C virus (HCV) affects approximately 58 million people globally [1], with 80% of those living with HCV residing in low- and middle-income countries (LMICs) [2]. Injection drug use (IDU) is the most common risk factor for HCV transmission [3] and an estimated 8.2 million people who inject drugs (PWID) around the world are HCV-antibody positive [4].

An estimated 35,000 PWID reside in Kenya [5,6]. Though HCV prevalence among PWID in Kenya and other parts of Sub-Saharan Africa (SSA) is low compared to the global prevalence [4,7], HCV incidence among PWID in this region is increasing. In Kenya, recent data estimate HCV prevalence is 1% in the western region, 13% in Nairobi, and 22% in the coastal region [8].

that the data contain potentially identifying and sensitive information that could compromise the privacy of research participants. Data inquiries can be sent to the Yale University Institutional Review Board at irb.support@yale.edu.

**Funding:** This work was supported by National Institutes of Health/National Institute on Drug Abuse (R01DA032080 to AK and PC; R01DA032080-05S1 to AK and PC; K99/R00DA043011 to MJA; and DP2DA053730 to MJA) and Gilead Sciences (IN-US-337-4656 to AM). The funders had no role in study design, data collection and analysis, decision to publish, or preparation of the manuscript.

**Competing interests:** The authors also confirm that there are no patents, products in development, or marketed products associated with this research to declare. Funding from Gilead Sciences does not alter our adherence to PLOS ONE policies on sharing data and materials.

The existence of safe and effective direct acting antiviral (DAA) medications provides great potential for HCV elimination, a World Health Organization target for 2030 [9], by providing a more easily tolerated course of treatment and expanding the scope of care [10,11]. However, gaps in diagnosis remain, and access to DAAs in many LMICs is limited [12,13]. Though evidence has demonstrated that PWID are motivated to undergo treatment and cure rates among PWID are comparable to other populations [14–16], treatment uptake remains limited, and care providers are often still hesitant to treat PWID [10]. Subsequently, many PWID living with HCV remain at risk of adverse health outcomes related to untreated HCV [17–19].

Reinfection after treatment also remains a concern. Though studies indicate similar sustained virologic response rates to non-PWID [20–23], PWID are at higher risk of reinfection due to ongoing risk behaviors [15,24]. Current research on reinfection rates in the DAA era is lacking, particularly for PWID in LMICs [10]. Research from Australia [25], Scotland [26], and Canada [24] indicates increasing reinfection rates among PWID since the scale-up of DAAs. This may be due to the fact that although not well-tolerated, interferon treatments required more interactions with healthcare providers and more opportunities for behavioral interventions [20,25,27]. Further investigation is needed as other analyses indicate comparable reinfection rates for both treatment types [15]. Other data examining factors related to reinfection highlight the key role of ongoing risk behaviors [15] and social environments [28] in shaping risk. Additionally, those engaged in medication-assisted treatment (MAT) [15] and engaged with harm reduction services [10] may be at lower risk of reinfection. Better understanding of reinfection risk factors among PWID in LMICs is needed to inform public health elimination efforts.

Given the low treatment uptake among PWID with HCV [29,30], unique barriers to treatment that affect PWID [16,31], and lack of data on reinfection [10], understanding the lived experience of PWID who have undergone treatment is key to informing intervention design and public health strategies. This is particularly true for LMICs, as treatment access in these settings remains more limited than in higher-income countries [12,13].

This study aimed to understand barriers in accessing HCV treatment and the impact of treatment on lifestyle and risk behaviors among PWID who were treated for HCV in methadone clinics and needle and syringe programs (NSPs)/drop-in centers (DICs) in Kenya. Given the lack of research capturing treatment experiences of PWID from LMICs with chronic HCV, these findings will contribute insight into implementing HCV treatment programs that meet this population's needs.

## Methods

### Study sample

This qualitative study was conducted from August 28 to September 27, 2019 as a follow-up study to the Testing and Linkage to Care for Injection Drug Users (TLC-IDU) study (R01DA0332080; NCT01557998) [8,32], which evaluated HCV and HIV prevalence and associated risk behaviors among PWID in Kenya. Participants in the parent study were recruited through respondent-driven sampling from July 15, 2015 to April 28, 2017. Those who received HCV treatment in the parent study were eligible to participate in this follow-up. The study aimed to reach as many participants as possible until saturation was reached.

### Study setting/procedure

Treatment procedures reflected standard practices and followed established viral hepatitis guidelines [33,34]. Participants received DAAs daily via directly-observed therapy at methadone clinics and NSP/DICs in Nairobi and coastal Kenya. The treatment was provided at no

cost. Those who traveled during the treatment period were offered take-home doses [34,35]. Some participants also had their DAAs brought to them by a DIC member on a case-by-case basis.

For those who were on methadone during HCV treatment, HCV treatment locations were sometimes co-located with methadone dispensation sites. In these cases, participants received their daily methadone dispensation before their HCV treatment; if the methadone appointment was missed, participants could not receive their DAAs for that day. Participants who received methadone and HCV treatment at different sites had to travel to both locations every day [34].

## Data collection

We purposively sampled participants of varied age and sex, and oversampled participants from Coastal Kenya due the higher HCV prevalence among PWID [8]. Because the parent study and treatment cohorts were comprised of mostly men [8,32,34] and as women in general make up a smaller proportion of the global PWID population [4,36,37], we also oversampled male participants. Overall, 36 individuals agreed to participate in the interviews. One interview was excluded from analysis due to missing data for the participant. No participants approached for this study refused to participate.

The interview guide included questions related to reinfection risk perceptions and behaviors, experiences receiving treatment (e.g., side effects, challenges or barriers, feelings after treatment), and current and previous substance use for clarity on participant's current risk behaviors. The interview guide was framed using Rhodes' Risk Environment Framework which focuses on the contexts in which behaviors occur and emphasizes orienting theory and research towards action, rather than focusing on individual action and responsibility [38,39]. This framework highlights the importance of examining lived experience to understand how sociocultural, structural, and other factors influence everyday practices related to harm and harm reduction and understands risk/risk behaviors as situated within a specific environment. The guide was discussed in team meetings with multidisciplinary team members (including nurse team leaders and research assistants) who spoke Swahili and were experienced in working with PWID. 'Transcreation' was used for the interview creation process to incorporate local and culturally relevant terms and to ensure regionally appropriate language was used [40]. The interview guide was then tested with peer case managers with IDU history for face validity in Nairobi and coastal Kenya.

Individual semi-structured interviews were conducted shortly after participants concluded HCV treatment by one of three female study team members (two research assistants—EK, AM and one program officer—MN) with diplomas in clinical medicine and social work and a bachelor's degree in economics (respectively) who were trained in open-ended interview techniques. Interviewers had familiarity and provider-client relationships with study participants from previous interactions through the parent study. One-on-one interviews were conducted with participants in private rooms at NSP/DICs in Nairobi and coastal Kenya and lasted approximately 45–60 minutes. Based on the responses from the participants, subsequent open-ended questions were asked for clarification. Interviewers took field notes regarding interview completion, reliability, and inconsistency in responses. Interviews were conducted in Swahili, audio recorded, transcribed in Swahili, and translated into English for data analysis (transcription done by all study team members in Kenya; translation done by MN).

## Data analysis

Data were analyzed using thematic analysis [41]. Four investigators (AM, HNM, LRR, MN) developed a coding guide to categorize themes that emerged from iterative readings of the

transcripts. Following this reading and initial coding, the codes were compared, classified into categories, and finally organized into themes. The research team then discussed the transcripts and guide, any coding discrepancies were resolved by consensus. The remaining transcripts were coded by four study team members (AM, HNM, LRR, MN). Once coded, we examined themes within and between codes to identify emergent themes. Each coder independently coded half of the transcripts in Dedoose (Version 9.0.107, Los Angeles, California), allowing for each to be coded by two investigators. The research team then discussed the coded interviews to refine common themes.

Findings have been reported in accordance with COREQ guidelines [42].

## Ethical considerations

This study was approved by the Ethics and Research Committee of Kenyatta National Hospital (University of Nairobi; IRB# P171/5/2011) and the Yale University Institutional Review Board (IRB# 1512016965). The study was conducted with support from the National AIDS and STI Control Program (NASCOP), local county-level government leaders, and local community leaders involved with work at the DICs and NSPs. All participants provided written informed consent. Participants were compensated 500 KSH. All cited excerpts are identified by the participant's study ID. Study IDs were not known to anyone outside of the research group.

## Results

### Demographics

From August 2019 to September 2019, 35 participants were interviewed at four DICs in Nairobi (n = 12) and five DICs in coastal Kenya (n = 23). Participants had a mean age of 38.2 years (SD = 6.5) and mostly male (n = 29, 82.9%) (Table 1). All participants had lifetime history of IDU, 14 reported using heroin via IDU at least once in the 30 days prior to the interview and 25 reported receiving methadone prescribed to them as part of a MAT program.

**Table 1. Participant demographic characteristics and substance use history.**

|  | Nairobi | | Coast | | Total | |
|---|---|---|---|---|---|---|
|  | N = 12 | % | N = 23 | % | N = 35 | % |
| Age, mean (SD) | 39.8 (7.91) | | 37.4 (13.8) | | 38.2 (6.45) | |
| **Sex** | | | | | | |
| Male | 11 | 91.7 | 18 | 78.3 | 29 | 82.8 |
| Female | 1 | 8.3 | 5 | 21.7 | 6 | 17.1 |
| **Relationship Status** | | | | | | |
| Single | 1 | 8.33 | 4 | 17.4 | 5 | 14.3 |
| In a relationship | 4 | 33.3 | 9 | 39.1 | 13 | 37.1 |
| Married | 3 | 25 | 6 | 26.1 | 9 | 25.7 |
| Previously married (divorced, separated, or widowed) | 4 | 33.3 | 4 | 17.4 | 8 | 22.9 |
| **Substance Use** | | | | | | |
| Ever injected | 12 | 100 | 23 | 100 | 35 | 100 |
| Ever accessed MAT | 9 | 75 | 16 | 69.6 | 25 | 71.4 |
| ***Used in Last 30 days*** | | | | | | |
| Heroin | 5 | 41.7 | 9 | 39.1 | 14 | 40 |
| Methadone (not prescribed) | 3 | 25 | 0 | 0 | 3 | 8.6 |
| Cocaine | 0 | 0 | 1 | 4.3 | 1 | 2.9 |
| Rohypnol/Bugizi | 0 | 0 | 1 | 4.3 | 1 | 2.9 |
| Marijuana | 3 | 25 | 4 | 17.4 | 7 | 20 |

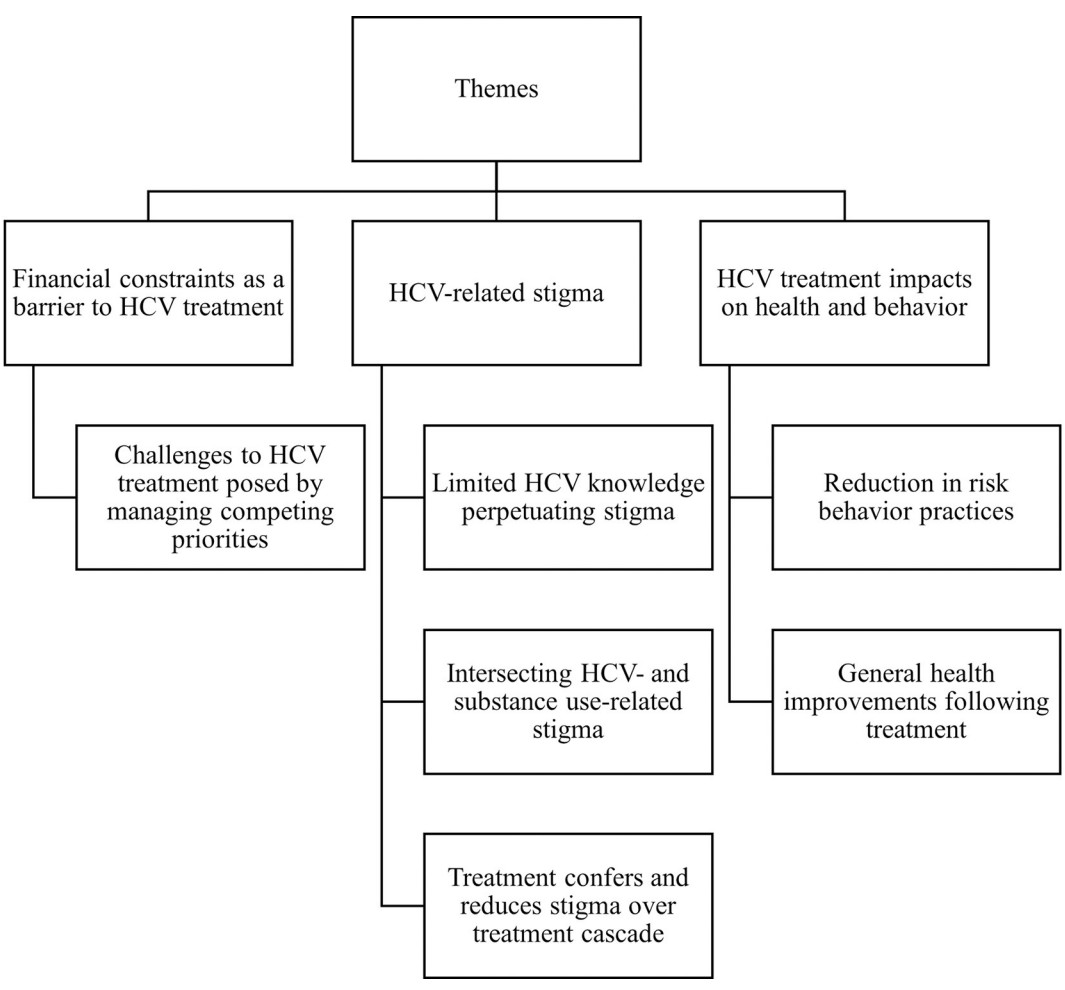

**Fig 1. Themes and subthemes.**

Our coding tree (Fig 1) included the following key themes: 1) Environmental constraints and stressors as barriers to HCV *treatment*; 2) HCV-related stigma; and 3) HCV Treatment impacts on health and behaviors.

## Financial constraints as a barrier to HCV treatment

During their interviews, participants frequently discussed financial factors, like inability to afford travel fare or food insecurity, as barriers to accessing and adhering to HCV treatment. Similarly, the need to manage other priorities connected with socioeconomic status like work, family, and MAT treatment was another barrier during HCV treatment.

The majority of participants traveled to an NSP/DIC or clinic each day to get their DAAs. Many participants discussed both financial constraints and distance as a significant barrier to treatment–often intertwined and dependent on treatment site and geographic location–particularly regarding affording fare to travel to get their medications.

*The major challenge I had is coming here every day. I could lack fare and sometimes I just did not have the morale to go. My morale was too low because of lack of employment to get fare from Dagoreti to Ngara [a 1-hour ride on public transit costing 100 Kenyan shillings]. (N/02/ 70640; Nairobi)*

Similarly, the need to travel to the hospital daily to take the medication posed some challenges:

*The first challenge that I was passing through was based on where I come from. And this thing [the medication] you could not be given. It was a must that you had to take it while at the hospital.* (C/06/00731; Coast)

Financial difficulties also made accessing and adhering to treatment difficult in other ways. Some participants said that they were told to take the medication with food, but that this was not always possible:

*The second thing was on the food side. Most of the time I used to take the medication while hungry I did not care that this may harm me more or whatever and we did not have food. It is my suggestion that someone would have been planned for food so that at least when one comes here before he is given the medicine he is given his food.* (C/06/00731; Coast)

Others spoke about the personal importance of the treatment to them and gratitude for the opportunity to be treated through the study; identifying socioeconomic and financial barriers that make accessing HCV treatment outside of a research study difficult.

*[HCV treatment] is very important. . . . Tell me where I could have gotten all that monies to buy those drugs. I thank those who donated. . . . our families are very poor that we could not afford that treatment.* (C/07/73423; Coast)

**Challenges to HCV treatment posed by managing competing priorities.** In addition to the financial and distance-related challenges to attending regular treatment visits, some participants discussed difficulty in managing the treatment schedule with other responsibilities such as work, family, and substance use treatment. For example, the strict appointment times for medication dispensation sometimes conflicted with this participant's work schedule:

*The challenge was with regards to treatment itself. We were being given instructions that we had to follow and it was sometimes a challenge when it comes to personal work because there was a special [appointment] time like 1000 hours and sometimes I could be tied up at work but had to leave work to get treatment depending on the time given by the doctor* (C/10/75411; Coast)

Another participant expressed a similar sentiment:

*You know I was still working even as I was taking my hepatitis medication so I had to rush and pick [up] the medication and then I run for work.* (C/08/73660; Coast)

Managing MAT and HCV treatment was an additional challenge for participants engaged in both treatment programs. Because methadone and HCV treatments were not always co-located or dispensed together, the additional fare and time spent traveling to multiple sites at different times presented an issue for some participants who were engaged in MAT:

*Finally the other challenge was that it was a must that I had to pass by the Hepatitis C treatment facility prior to going for my treatment at [the methadone clinic]. And it was every day.* (N/02/70402; Nairobi)

For those whose methadone and HCV treatments were co-located, the requirement to take HCV treatment at the same time as methadone presented a barrier to HCV treatment access and adherence:

*Also hepatitis was linked to the Methadone daily dose so when I miss methadone and am suspended it means that I also miss the hepatitis medicines. But I did not miss for more than three consecutive days.* (*N/04/71421; Nairobi*)

## HCV-related stigma

Participants discussed experiencing HCV-related stigma prior to and during treatment, but also subthemes of how limited HCV knowledge among PWID and other community members contributed to negative attitudes toward HCV, how HCV- and substance use-related stigmas intersect, and the ways receiving HCV treatment affected their perceptions of stigma.

**Limited HCV knowledge perpetuating stigma.**   Some experiences of stigma described by participants resulted from associations related to illness and others' fears of contracting HCV, suggesting a lack of knowledge about HCV transmission and treatment/cure:

*Ahh at first when people heard that I had hepatitis they thought that I would infect them to some extent even sharing a cup of water was a problem even walking or sitting together in one place. . . Most people did not believe the disease could be healed it was very difficult for them to believe but now they believe and we are now best of friends.* (*N/02/70640; Nairobi*)

Another participant from the Coastal region had a similar experience, also reflecting on how associations between substance use and HCV contributed to others' negative perceptions of them:

*Initially people used to say that this disease does not have cure and it is a disease that is only prevalent among the drug users . . . Initially they used to fear interacting and associating with me. There was a lady that tested us and promised to bring treatment and when we got the treatment slowly people started trusting us* (*C/09/74226; Coast*)

Others reported experiencing stigma as a result of others mistaking DAAs for antiretrovirals (ARVs) believing they had HIV.

*One of the challenges that I experienced while here on medication was like this, our peers whom we take methadone with when the time comes to take this treatment madam used to out the medicine like this (shows how) now our peers did not understand whether we were taking hepatitis C drugs they thought it was ARVs they did not know if I had hepatitis or I had HIV because they remained questioning and talk about me a lot.* (*C/09/74240; Coast*)

**Intersecting HCV- and substance use-related stigma.**   Some participants discussed stigma related to HCV and substance use. For example, this participant reported that despite not disclosing their HCV or treatment status they were still regarded negatively by community members as a result of stigma toward substance use:

*. . . they still perceive me as an addict [despite being treated for HCV]. . . For the hepatitis C not so many people know that I had it so the [stigma] was only from the fact that I was an addict and hence they feared that we addicts tend to steal things and sell.* (*C/07/00737; Coast*)

**Treatment confers and reduces stigma over the treatment cascade.** In spite of the substantial stigma they did experience, some participants reflected on the stigma members of their community had towards them prior to treatment and how this dissipated upon completing treatment:

*Yes. . . . It has changed my life. However I can say. . . I mean we are now healed . . . There was a lot of stigma during the initial stages because there are so many hepatitis, there are some people who did not want even to greet others but since we got the treatment for now they take it like it is something that we were treated and got healed.* (C/06/00731; Coast)

Similarly, participants reported that they experienced less stigma from other PWID after completing treatment:

*It has changed [stigma from others]. At first someone would look at you and maybe they need your help to inject them and they are afraid that when you touch them you will infect them but for now they are not afraid because they know that you have been treated and hence can help each other.* (C/10/00757; Coast)

## HCV treatment impacts on health and behavior

Participants were asked to discuss any changes that resulted from treatment in their lives and health, as well as in their substance use and sexual behaviors. The discussions of these different changes demonstrate how HCV treatment impacted participants' overall health and risk behaviors.

**Reduction in risk behavior practices.** The majority of participants stated that being treated for HCV reduced their risk behavior practices (e.g., stopping IDU, stopping all drug use completely, or no longer sharing needles/syringes), such as this participant:

*I changed a lot. I stopped sharing needles. In fact, I totally stopped injection and now I only smoke. . . injecting had resulted to the collapse of my veins. . . . With regards to sex, I totally stopped. You know the doctor told me to stop many things till I get well . . .* (N/04/71421; Nairobi)

Similarly, several others discussed changes they made to their sexual practices, including using condoms during sex and asking potential partners about their status:

*It has changed because if I want to have sex I must use protection and that protection protects one from getting the virus and also if you do not know your partner it is a must you both go for testing for the partner to know their status and all of you to know how you are to agree with each other* (N/01/00702; Nairobi)

For a some, fear of being reinfected was mentioned as a driver of behavior change:

*It [being treated] changed my use of illicit drugs. . . . Because I was afraid since the doctor told me that if I continue injecting that is how the disease will come back to me and hence I stopped injecting.* (C/06/72655; Coast)

**Improvements in overall health following HCV treatment.** Some participants discussed other impacts of treatment, such as motivation to live a generally healthier life:

*I have seen changes let us say like showering I never used to shower and even my clothes I never used to change but nowadays I see I am very clean I can shower and change clothes and I feel God is indeed great.* (N/04/71625; Nairobi)

*The changes that I have are for now I am too careful with myself. I make sure that I eat a healthy diet* (N/04/68904; Nairobi)

Several participants referenced specific physical health improvements after treatment, such as reductions in pain/discomfort, improvements in energy levels and appetite, and weight gain:

*I am glad that my health is no longer as it used to be. I had deteriorated. The second thing, on my body I can see I am going on well and also there is a time when I could take the medicine and feel pain, little did I know that it is my liver that had been infected. For now, when I use the medication it does not take time to act if it is fever it just ends. I am grateful that I can see the function of my liver is back to normal.* (C/06/00731; Coast)

Others did not mention specific symptoms that had resolved after treatment, but spoke more generally about the interplay of feeling healthy with greater productivity:

*Other changes are on my body i.e. healthy body i.e. you got energy that can work. You don't look like you are so weak you have morale to work not like initially. . . but for now I feel there are great changes because I work and I earn my own living and be able to eat.* (N/04/71625; Nairobi)

## Discussion

Our data from PWID who received HCV treatment in two regions in Kenya provide valuable insights into patients' perceptions of barriers to HCV treatment. Such lived experience can shed light on how individual and structural factors (e.g., patients' socioeconomic status and structural factors like treatment locations and transportation) impact treatment and may guide design and implementation of HCV treatment interventions that meet the needs of PWID in LMICs. In particular, participants' discussions of barriers to care and stigma they experienced gives insight into the context PWID considering or seeking HCV treatment are situated within, and how these factors may affect likelihood of treatment adherence and continuation of risk behaviors. These interviews also provide support for the benefit of treating PWID and other high-risk populations that may be engaged in ongoing risk behaviors at the start of or throughout treatment.

Environmental stressors, financial challenges, and competing priorities are well-established barriers to accessing HCV treatment in both LMICs and higher-income settings [29,31,43–45]. In our interviews, participants' discussions of the challenges they experienced during the treatment process are particularly useful for guiding future direction of treatment interventions. Given the financial challenges associated with traveling to treatments and obtaining adequate meals to take with the medication, intervention designs for treatment that is not co-located with other services should consider incorporating transportation to and from clinics and providing nutritional support to those experiencing food insecurity. Studies show that food insecurity is a barrier to HIV treatment adherence and viral suppression [46,47] and interventions that addressed this issue improved ARV adherence and overall quality of life [48–50]. Similarly, studies from higher-income countries demonstrate that providing

transportation to treatment appointments reduces financial burden for PWID engaged in HCV treatment [51]. Other treatment programs for HIV in resource-limited settings provide transportation stipends or reimbursements to patients [52], which also helps to reduce financial obstacles to care and may be more feasible to implement logistically than directly providing transportation. Additionally, allowing for flexibility in scheduling and medication dispensation (e.g., offering the options for take-home doses) would further alleviate patients' burdens and make treatment more accessible to those who must balance several priorities (e.g., work, childcare, other healthcare appointments) with their HCV care; in other studies, increased flexibility in appointment scheduling enabled more successful care [51,53]. In addition to these financial challenges, participants also noted that they would not have been able to afford the treatment were it not provided for free through the study. The additional financial burden of treatment costs is likely to pose another challenge to access and treatment adherence for those who must cover the cost of their own treatment [54,55]. Given associations between financial stress and substance use [56], the potential associated financial strain may also affect the likelihood of continuation of risk behaviors in PWID covering their treatment cost.

Participants' challenges managing both methadone and HCV treatment schedules indicate the potential benefits of integrating these treatment programs and offering MAT at HCV treatment centers (and vice versa), as well as allowing for greater flexibility in scheduling policies to better accommodate the various priorities that patients must balance. Hospital-based treatment has been shown to be less ideal for PWID as it often increases the distance required to travel for care [16], so co-locating substance use and HCV treatment would not only provide more opportunities for engagement with both types of treatment, but would also reduce a common barrier to treatment. Previous studies conducted in both high- and low-resource settings have shown that co-location of these services is effective [57–61], and PWID interviewed prior to initiating HCV treatment supported integrating DAA dispensation into DICs or MAT programs as they are already visiting these places (i.e., co-location would reduce need for additional travel time and expense) [31]. Additionally, another cohort of Kenyan PWID living with HIV found those participating in a methadone treatment program were more likely to be on ART and to have achieved viral suppression [62]; for participants of this study, co-location of treatment mitigated challenges in accessing treatment.

In the year following these interviews, the COVID-19 pandemic and subsequent social distancing practices and lockdowns had significant effects on healthcare, substance use treatment, and harm reduction service provision in many countries [63–67]. For example, near the start of the COVID-19 pandemic, the DICs and MAT clinics involved in this study reduced the number of services provided and clients served, which lead to new challenges in performing psychosocial and outreach services [63]. The decentralization of some services provided the opportunity to improve certain services and expand reach. For example, some opened satellite clinics in new locations and others sent peer educators to distribute resources (e.g., meals, condoms, etc.) directly to clients' homes. Participants in this study emphasized the importance of these services and suggested that more flexibility in treatment options helped address the challenges they faced in engaging in HCV treatment, so it is likely that these COVID-19-related service provision changes would be particularly beneficial in reducing barriers to care for future patients [63].

Participants' experiences with stigma are consistent with existing literature showing high levels of stigma around HCV and its negative effects on health outcomes [31,43,45,68,69]. HCV stigma has been associated with negative perceptions of IDU [68,70,71], also reflected in participants' discussions. The stigma that many participants reported experiencing has important implications for care engagement and wellbeing, as perceived stigma has also been

associated with decreased engagement in care [16,31,69,70]. Group treatment to address patients' social support needs may help to reduce the effects of stigma on care engagement [31,72].

Participants' reports of stigma related to others' beliefs that they were receiving treatment for HIV/AIDS suggest that greater community-level HIV and HCV education is needed and may also reduce stigma experienced by those seeking treatment. Similarly, community members' misunderstanding of how HCV is transmitted (e.g., not wanting to be physically close to and not wanting to share cups with those who have HCV) also show a need for greater education among the general population.

Previous literature has demonstrated lack of knowledge as a significant barrier to care [29,43,45], also supports this as an important step for preventing future infections and improving care engagement. Participants' discussions of changing their risk behaviors or following treatment instructions as a result of influence from medical providers also indicates that better understanding of HCV transmission and treatment is beneficial to patients and may lead to better treatment adherence and motivate patients to change their risk behaviors. Similarly, this lack of understanding also points to the importance of peer educators as sources of information whose lived experience may be beneficial to those considering/undergoing treatment.

Reports of changes in risk behaviors in these interviews are consistent with other research showing reductions in injecting and needle/syringe sharing during and after HCV treatment with both interferon and DAAs in higher-income settings [70,73–75]. Prior to initiating DAA treatment PWID from the parent study frequently stated that they would refrain from engaging in risk behaviors once cured [31]. Potential explanations for reductions in risk behavior include more frequent interactions with healthcare providers and opportunities for behavioral health interventions and risk education [20], improvements in self-efficacy and reduction in feelings of shame [69,76], and a change in self-concept as related to disease status [77]. Given the role that ongoing risk behaviors play in reinfection [15], these data also underscore the importance of providing patients with additional support for behavioral changes, such as linkage to MAT programs, NSPs, and more in-depth HCV risk education.

## Limitations

One limitation of this study is the potential for desirability bias in participants' responses because interviews were conducted after treatment, though many participants' candidness about their ongoing risk behaviors indicate that this may not have been a significant limiting factor. This is likely mitigated by the fact that the interviewers were not the ones directly involved in providing treatment (i.e., were not the clinicians providing care or dispensing medication). Most of the participants in this study were also either currently or previously engaged in MAT treatment, and as a result this sample may be less representative of all PWID seeking or undergoing HCV treatment as a whole. Additionally, as the majority of our study participants were male, we may not have fully captured the experiences and unique challenges experienced by women who inject drugs and seek HCV treatment. As women who inject drugs have been shown to face challenges related to gender roles, discrimination, and economic inequality [78], more in-depth of how these factors affect HCV treatment access for women is needed. Participants in this study also expressed gratitude for the ability to receive treatment for free through the study, as the cost of treatment would have otherwise been prohibitive. It is possible that impacts of treatment on lifestyle and risk behaviors may differ for individuals paying high treatment costs, as the financial burden may impact stress levels and treatment adherence.

## Conclusions

This study investigates the experiences of PWID receiving DAA treatment for HCV in Kenya. The findings can be used to improve HCV treatment programs in SSA to ensure that they meet the needs of this uniquely vulnerable population. As treatment access in SSA and uptake within PWID populations has been limited, the conclusions gleaned from these findings may be valuable in reducing these disparities and preventing poor health outcomes caused by untreated HCV. Given both the high risk of reinfection for the population, as well as the overall lack of data on reinfection in the DAA era, application of these findings to improve the effectiveness and accessibility of treatment is particularly important.

## Supporting information

**S1 Checklist.**
(DOCX)

## Acknowledgments

We thank the following deeply for their contributions and support: Peer Case Managers, the Kenya Ministry of Health, the NSP/DIC and MAT sites that provided treatment, and the study participants.

## Author Contributions

**Conceptualization:** Matthew J. Akiyama, Peter Cherutich, Ann Kurth, Abbe Muller.

**Data curation:** Hannah N. Manley, Lindsey R. Riback, Mercy Nyakowa, Abbe Muller.

**Formal analysis:** Hannah N. Manley, Lindsey R. Riback, Mercy Nyakowa, Abbe Muller.

**Funding acquisition:** Matthew J. Akiyama, Peter Cherutich, Ann Kurth, Abbe Muller.

**Investigation:** Mercy Nyakowa, Abbe Muller.

**Methodology:** Matthew J. Akiyama, Peter Cherutich, John Lizcano, Ann Kurth, Abbe Muller.

**Project administration:** Matthew J. Akiyama, Peter Cherutich, John Lizcano, Ann Kurth, Abbe Muller.

**Resources:** Matthew J. Akiyama, Peter Cherutich, John Lizcano, Ann Kurth.

**Supervision:** Matthew J. Akiyama, Peter Cherutich, John Lizcano, Ann Kurth.

**Visualization:** Hannah N. Manley.

**Writing – original draft:** Hannah N. Manley, Lindsey R. Riback, Mercy Nyakowa, Abbe Muller.

**Writing – review & editing:** Hannah N. Manley, Lindsey R. Riback, Mercy Nyakowa, Matthew J. Akiyama, Peter Cherutich, John Lizcano, Abbe Muller.

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
