## [Decision Letter · Decision Letter 0]

13 Aug 2024

PGPH-D-24-00981

Hepatitis C treatment experiences and risk behavior changes among people who inject drugs in Kenya: a qualitative study

Dear Dr. Muller,

Thank you for submitting your manuscript to PLOS Global Public Health. After careful consideration, we feel that it has merit but does not fully meet PLOS Global Public Health’s publication criteria as it currently stands. Therefore, we invite you to submit a revised version of the manuscript that addresses the points raised during the review process.

Academic Editor Guillaume Fontaine's comments: Thank you for submitting your work for our consideration. Enclosed you will find the peer reviewer reports. While there is interest in your paper, some areas require revision. Reviewer 1 has noted that the results section needs better alignment with the study’s aims. The study intends to explore treatment experiences and risk factors, yet the results primarily focus on barriers to HCV treatment and the impact of HCV treatment on individuals’ lifestyles and risk-taking behaviors, without addressing the personal experiences of receiving treatment. It would be beneficial to revise the title and aim of the paper to address this. Reviewer 2 pointed out that the study engaged few women, and further elaboration on the potential reasons and implications is necessary. This aspect should also be acknowledged in the study’s limitations. Additionally, while the study is guided by Rhodes’ Risk Environment Framework, few details are provided on this framework, and it is not referenced in the discussion to contextualize the findings. I recommend expanding on this framework and integrating it into your discussion. Finally, please provide the completed COREQ checklist as supplementary material.

We look forward to receiving your revised manuscript.

Kind regards,

Guillaume Fontaine, PhD, RN

Academic Editor

Journal Requirements:

2. The resolution of Figures 1 is very low and somewhat difficult to read. It is important that our Editors and Peer Reviewers are able to read all parts of a submission. Please replace these figures with higher resolution copies.

Additional Editor Comments (if provided):

N/A

Reviewers' comments:

Reviewer's Responses to Questions

**Comments to the Author**

1. Does this manuscript meet PLOS Global Public Health’s publication criteria? Is the manuscript technically sound, and do the data support the conclusions? The manuscript must describe methodologically and ethically rigorous research with conclusions that are appropriately drawn based on the data presented.

Reviewer #1: Yes

Reviewer #2: Yes

Reviewer #3: Yes

2. Has the statistical analysis been performed appropriately and rigorously?

Reviewer #1: Yes

Reviewer #2: Yes

Reviewer #3: N/A

3. Have the authors made all data underlying the findings in their manuscript fully available (please refer to the Data Availability Statement at the start of the manuscript PDF file)?

Reviewer #1: Yes

Reviewer #2: Yes

Reviewer #3: No

4. Is the manuscript presented in an intelligible fashion and written in standard English?

Reviewer #1: Yes

Reviewer #2: Yes

Reviewer #3: Yes

5. Review Comments to the Author

Reviewer #1: Thanks for this opportunity to review this manuscript. This study explored the experiences of PWID with Hepatitis C treatment and the changes in their risk-taking behaviors after treatment in Kenya. The authors applied robust methods and provided valuable insights into some of the barriers to treatment which have the potential to be used for designing future interventions. Below, please see my comments and suggestions.

Methods

1) Study sample-The study participants were recruited among those who received treatment in the parent study. Could you please clarify whether the HCV treatment procedures/practices and follow ups in the parent study were different from the usual practices (i.e. how HCV treatment is normally offered at those centers)? (Please discuss this in Methods section)

• Is there any concern that experiences of those who received treatment via the parent study might be different from those who are treated via other routes (usual practices)? Please discuss its implications and how it can affect study findings. (Please discuss this in Discussion)

2) Please provide more details on your sample size; how did you decide on the number needed? how many individuals did you ask to participate and how many refused to participate?

Results

3) I suggest removing the age and gender of participants from identifiers after quotes and simply assign a number to them. I am worried that participants could be identified using age and gender, especially since they are frequenters of harm reduction centers.

4) I suggest using “Environmental constraints/stressors and competing priorities” instead of the theme “Financial constraints”, because in addition to financial problems, you also discuss difficulty travelling to the treatment center, affording food, and managing other priorities.

5) Please add (in this section or in the methods) that all the participants received treatment for free. Also, in your discussion, please explain whether HCV treatment is covered in Kenya and how that could impact the generalizability of your findings to those who receive treatment through usual practices.

6) Line 204- “In addition to the financial and geographic barriers to attending treatment sessions, participants discussed the challenges of navigating substance use recovery, stigma, and the role of family support (or lack thereof) that arose during HCV treatment” fits better under “HCV related stigma”.

7) As mentioned by the authors, the study aims to understand treatment experiences and risk factors of PWID. However, the results do not discuss individuals’ experiences of receiving treatment itself, for example did they experience any side effects, how was the interaction with the staff and healthcare providers, feelings about treatment, …. Instead, the results focus on barriers to HCV treatment and impact of HCV treatment on individuals’ lifestyle/risk taking behaviors. I believe the title and aim of the study should reflect these results. I suggest modifying the title and aim slightly to something similar as this: “barriers to/challenges of (receiving) Hepatitis C treatment and the impact of treatment on risk-takings behaviors among people who inject drugs in Kenya: a qualitative study”.

Discussion

8) Line 327- “Our data from PWID who received HCV treatment in two regions in Kenya provide valuable insights into patients’ perceptions about the HCV treatment process.” I find this sentence a bit misleading as the treatment process is not discussed in the results. As mentioned previously, I suggest this: “Our data from PWID who received HCV treatment in two regions in Kenya provide valuable insights into patients’ perceived barriers to HCV treatment…”

9) Line 332- “They also provide support for the feasibility and benefit of treating PWID”, although the findings support the benefits of treating PWID, they do not show whether treatment is feasible. I would remove “feasibility”.

10) In discussion, you mention the effect of the pandemic on individuals’ experiences with HCV treatment. Please report this finding in your results as well and provide some quotes from participants if possible.

11) Line 392- “The lack of HCV knowledge demonstrated by some participants, in conjunction with previous literature demonstrating lack of knowledge as a significant barrier to care.” The authors discuss lack of HCV knowledge as a significant barrier to care. However, the results only discuss lack of knowledge of others (not the participants themselves, but their friends, family…) as a factor contributing to stigma. The results do not discuss participants’ knowledge with regards to HCV (or HCV treatment) and its implications, as a result it is not possible to conclude that lack of knowledge of participants was a major barrier.

Reviewer #2: This is an interesting and important research study which highlights the views and lived experience of people who inject drugs in Kenya regarding barriers to treatment for hepatitis C.

The research connected well with the existing literature in terms of barriers around community understanding of HIV and HCV, as well as stigma and changing risk behaviours. The manuscript is well written and justifies itself in light of the increasing prevalence of HCV among PWID in the region. The data analysis & coding process appears to be rigorous.

I noticed that the participant sample skews heavily towards men and I felt this throughout the paper. In light of this, I found myself wanting a little bit more background on gendered roles in Kenya among men and women, at work and with family. As a reader I don't know anything about men and women in Kenya and how they might be affected differently by stigma, finances, work, and responsibilities around home and family.

Given that this is a paper about the lived experience of at-risk populations, it feels important to perhaps acknowledge and explain the fewer women in the sample, why that might be, and potentially include this as a recommendation for future research to take into account. Are there fewer female PWID in Kenya, are women less able or willing to talk to researchers, or something else? If all of this is outside the scope of the paper, it might be helpful to acknowledge this in the limitations section.

In terms of the conclusion, it may be difficult to claim that the findings can inform improvements to treatment in other LMICs, given the range of countries fitting this definition. While it's probably reasonable to assume that some of the findings could be applied elsewhere, I think that providing some more detail on the similarities between Kenya and other LMICs in terms of the research findings, or limiting this claim to Sub-Saharan Africa for example, would strengthen this statement.

Overall I really enjoyed the manuscript and respect the work that has been put into it.

Reviewer #3: Thank you for the opportunity to review this manuscript regarding HCV care engagement among people with a history of injecting drug use in two regions in Kenya. This is a well written paper and I agree with the authors that this paper provides important insights for improving HCV treatment programs in LMICs. The Risk Environment Framework aligns well with the research.

I have made minor comments (mostly editing suggestions) for the authors' consideration.

Abstract -

Spell out LMICs in first instance.

Mention the theoretical framework which informed the analysis.

Keywords - consider changing "qualitative" to "qualitative research"

Line 75 - should "program" be plural?

Line 147 - suggest clarifying if this is illicit or prescribed methadone use

Lines 155-157 - How were these determined/identified?

Lines 168-170 (quote) - Could the authors provide context of how far this is?

It is meaningful for the reader to understand the challenge, eg., more than 1 hour by bus

Line 202 (quote) - Should this be "pick up the medication"?

Line 219 (Mathare) - I had to Google this. Perhaps some context for the reader?

Line 236 (quote) - "when people had that I" - Should "had" be 'heard'?

Lines 238-239 and 250 -257 - The two quotes presented align - the stigma and fear of people living with HCV, but also the lack of public knowledge that HCV is now treatable (which seems to further compound experiences of stigma among people receiving HCV treatment). Would it be worthwhile to link the two quotes and expand to mention lack of knowledge re cure?

Line 258 "still" - delete - this appears to be an extra word inserted.

Line 262 "the treatment" - Are they referring to HCV treatment here or MAT? It isn't clear how people know he uses drugs if they didn't know he had hep C.

Line 276 "treatment" - Should this be "stigma" instead of "treatment" for reader clarity about what is being referred to?

Line 328 "HCV treatment process" - and challenges of access?

Lines 344 - 346 - Is it being suggested that transport for HCV treatment patients should be provided in Kenya? Is this a feasible consideration?

6. PLOS authors have the option to publish the peer review history of their article (what does this mean?). If published, this will include your full peer review and any attached files.

**Do you want your identity to be public for this peer review?** For information about this choice, including consent withdrawal, please see our Privacy Policy.

Reviewer #1: No

Reviewer #2: **Yes: **Thomas Rudge

Reviewer #3: No

---

## [Editor Report · Decision Letter 1]

12 Dec 2024

Barriers to and impacts of Hepatitis C treatment among people who inject drugs in Kenya: a qualitative study

PGPH-D-24-00981R1

Dear Dr. Muller,

We are pleased to inform you that your manuscript 'Barriers to and impacts of Hepatitis C treatment among people who inject drugs in Kenya: a qualitative study' has been provisionally accepted for publication in PLOS Global Public Health.

Best regards,

Guillaume Fontaine, PhD, RN

Academic Editor